# Preparation and Characterization of Transparent Polyimide Nanocomposite Films with Potential Applications as Spacecraft Antenna Substrates with Low Dielectric Features and Good Sustainability in Atomic-Oxygen Environments

**DOI:** 10.3390/nano11081886

**Published:** 2021-07-23

**Authors:** Yan Zhang, Bo-han Wu, Han-li Wang, Hao Wu, Yuan-cheng An, Xin-xin Zhi, Jin-gang Liu

**Affiliations:** 1Beijing Key Laboratory of Materials Utilization of Nonmetallic Minerals and Solid Wastes, National Laboratory of Mineral Materials, School of Materials Science and Technology, China University of Geosciences, Beijing 100083, China; 3003200016@cugb.edu.cn (Y.Z.); 2003200021@cugb.edu.cn (H.W.); 2103190039@cugb.edu.cn (Y.-c.A.); 3003200015@cugb.edu.cn (X.-x.Z.); 2Space Materials and Structure Protection Division, Beijing Institute of Spacecraft Environment Engineering, Beijing 100094, China; wubohan@spacechina.com; 3Shandong Huaxia Shenzhou New Material Co., Ltd., Zibo 256401, China; whl89333@huaxiashenzhou.com

**Keywords:** polyimide substrate, atomic oxygen, dielectric constant, POSS, antenna

## Abstract

Optically transparent polyimide (PI) films with good dielectric properties and long-term sustainability in atomic-oxygen (AO) environments have been highly desired as antenna substrates in low earth orbit (LEO) aerospace applications. However, PI substrates with low dielectric constant (low-*D*_k_), low dielectric dissipation factor (low-*D*_f_) and high AO resistance have rarely been reported due to the difficulties in achieving both high AO survivability and good dielectric parameters simultaneously. In the present work, an intrinsically low-*D*_k_ and low-*D*_f_ optically transparent PI film matrix, poly[4,4′-(hexafluoroisopropylidene)diphthalic anhydride-*co*-2,2-bis(4-(4-aminophenoxy)phenyl)hexafluoropropane] (6FPI) was combined with a nanocage trisilanolphenyl polyhedral oligomeric silsesquioxane (TSP-POSS) additive in order to afford novel organic–inorganic nanocomposite films with enhanced AO-resistant properties and reduced dielectric parameters. The derived 6FPI/POSS films exhibited the *D*_k_ and *D*_f_ values as low as 2.52 and 0.006 at the frequency of 1 MHz, respectively. Meanwhile, the composite films showed good AO resistance with the erosion yield as low as 4.0 × 10^−25^ cm^3^/atom at the exposure flux of 4.02 × 10^20^ atom/cm^2^, which decreased by nearly one order of magnitude compared with the value of 3.0 × 10^−24^ cm^3^/atom of the standard PI-ref Kapton^®^ film.

## 1. Introduction

Flexible antennas play important roles in low-earth orbit (LEO) spacecraft for space-earth communications, data transmission, and navigation [1]. In most cases, the spaceborne antennas will contact the LEO environments in the whole life of the spacecraft. The severe space environments in LEO, including atomic oxygen (AO), thermal shock cycle in the range of −120 °C~150 °C, ionizing radiation, micrometeoroids, and so on will inevitably deteriorate the antenna materials, especially the common light-weighted polymeric or polymer composite antennas [2,3,4,5]. When the spacecraft cruised in LEO at the speed of 7.9 km/s, the impact energy between the collision of polymer antennas and the AO species might be as high as 5 eV. This usually causes severe corrosion of the polymer substrates in the antennas due to the dissociation of chemical bonds in the polymers by the high-energy impacts [6,7,8,9,10]. For example, for the common polyimide (PI) substrates derived from pyromellitic anhydride (PMDA) and 4,4′-oxydianline (ODA) (PI_PMDA-ODA_), which are most widely used at present in spacecraft antennas, as shown in Figure 1, the molecular structure is mainly composed of C-N bond (dissociation energy: 3.2 eV), C_6_H_5_-H bond (dissociation energy: 4.8 eV), and -C_6_H_5_-C(=O)- bond (dissociation energy: 3.9 eV) [11]. Thus, serious corrosion might occur when the PI_PMDA-ODA_ substrates were impacted by AO, which might greatly decrease the servicing life of the antenna [12]. The poor AO resistance of standard PI_PMDA-ODA_ substrates undoubtedly prohibited the applications in LEO spacecraft antennas. On the other hand, for the antenna applications, low dielectric constant (low-*D*_k_) and low dielectric dissipation factor (low-*D*_f_) features are often required for the polymeric substrates in order to achieve a high signal transmission speed (low signal delay) and low crosstalk [13,14]. In addition, the low dielectric properties could also provide the antenna with better impedance matching and higher gain compared to conventional high-dielectric-constant substrates [15]. The common PI antenna substrates PI_PMDA-ODA_ exhibited a *D*_k_ and *D*_f_ value of 3.2 and 0.02 at 1 MHz frequency, respectively [16,17,18]. Thus, either the poor AO resistance or the high dielectric properties of the current PI_PMDA-ODA_ substrates could not meet the property requirements of advanced antennas for LEO spacecraft.

Recently, Meador and coworkers reported the pioneering work on the development of ultra-low-*D*_k_ PI aerogels as substrates for lightweight patch antennas for potential aerospace applications [19,20]. Relatively low *D*_k_ values as low as 1.16 were obtained from PI aerogels prepared by cross-linking 3,3′,4,4′-biphenylenetetracarboxylic dianhydride (BPDA) and 2,2′-dimethylbenzidine (DMBZ) with 1,3,5-triaminophenoxybenzene (TAB). It could be deduced from the molecular structures of the derived PI aerogels that the antennas made from the BPDA-DMBZ-TAB system might be sensitive to AO exposure and the applications of such antennas in LEO spacecraft might be of great challenge. In our previous work, we developed a series of AO-resistant PI aerogels for the potential antenna applications from BPDA, DMBZ, an aromatic diamine containing pendant polyhedral oligomeric silsesquioxane (POSS) units, and a POSS-containing crosslinker, octa(aminophenyl)silsesquioxane (OAPS) [21]. The inorganic-organic hybrid POSS components endowed the PI aerogels with excellent AO resistance. However, the dielectric properties of the POSS-PI aerogels were not mentioned.

In the current work, in order to remedy the poor AO resistance and relatively high *D*_k_ values of the common PI antenna substrates, a series of PI composite substrates were developed with fluoro-containing PI as matrix and trisilanolphenyl-POSS (TSP-POSS) as additive. First, the incorporation of fluorinated blocks has been well established to be able to decrease the *D*_k_ values of the PIs due to the low molar polarizability or high electronegativity of fluoro-containing substituents [22]. Secondly, the incorporation of POSS, especially the highly polar TSP-POSS additives could efficiently increase the AO-resistant properties of the PI composites [23]. Thirdly, the nano-caged nature of POSS could decrease the density of the PI composite films via the confined air, thus could further decrease the *D*_k_ values of the PI composites [24]. The effect of the TSP-POSS additive on the comprehensive of PI composite substrates, especially on the dielectric properties and AO erosion properties were investigated in detail.

## 2. Materials and Methods

### 2.1. Materials

Fluoro-containing monomers, including 4,4′-(hexafluoroisopropylidene)diphthalic anhydride (6FDA) and 2,2-bis[4-(4-aminophenoxy)phenyl]hexafluoropropane (BDAF) were all purchased from Daikin Chemicals (Tokyo, Japan). The dianhydride 6FDA was pretreated in a vacuum at 180 °C for 10 h, while the diamine BDAF was used without further treatment. Trisilanolphenyl-POSS (TSP-POSS) was purchased from Hybrid Plastics, Co., Ltd. (Hattiesburg, MS, USA). The ultra-dry *N*,*N*-dimethylacetamide (DMAc) was purchased from Beijing Innochem Science & Technology Co., Ltd. (Beijing, China). The other reagent-grade chemicals were all purchased from Phentex (Tianjin) Fine Chemicals Co., Ltd. (Tianjin, China).

### 2.2. Measurements

A Brookfield DV-II+ Pro viscometer (Ametek, Middleboro, MA, USA) was used to measure the absolute viscosities of the poly(amic acid) (PAA) solutions, and the temperature was controlled at 25 °C. Attenuated total reflectance Fourier transform infrared (ATR-FTIR) spectra were recorded on a Bruker Tensor-27 FT-IR spectrometer (Ettlingen, Germany) at a resolution of 2 cm^−1^. Wide-angle X-ray diffraction (XRD) was obtained on a Bruker D8 advanceX diffractometer (Ettlingen, Germany) using Cu-Kα1 radiation. An SCALab220i-XL electron spectrometer (Thermo Fisher Scientific, Waltham, MA, USA) was used to obtain the X-ray photoelectron spectroscopy (XPS), under the monochromatic MgKα radiation. The surface morphology was obtained on a Technex Lab Tiny-SEM 1540 field-emission scanning electron microscope (FE-SEM) (Tokyo, Japan) and a Bruker Multimode 8 atomic force microscopy (AFM) microscope (Santa Babara, CA, USA).

The optical transmittance of the films was measured on a Hitachi U-3210 Ultraviolet-visible (UV-Vis) spectrophotometer (Hitachi, Ltd., Tokyo, Japan). The color parameters including yellow index (YI) and haze of the PI films at a thickness of 25 µm were measured by an X-rite color i7 spectrophotometer (Grand Rapids, MI, USA). Among the obtained color parameters, the meaning of each parameter is as follows: *L** is the lightness, where 100 implies white and 0 indicates black; *a**: positive value means red, negative means green; *b**: positive value was yellow, negative was blue. A Metricon MODEL 2010/M prism coupler (Pennington, NJ, USA) was used to measure the refractive index of the PI film at the wavelength of 1310 nm. The average refractive index (*n*_AV_) was calculated according to Equation (1):(1)$$n_{\mathrm{AV}}=\left(2n_{\mathrm{TE}}+n_{\mathrm{TM}}\right)/3$$
where, *n*_TE_ was the in-plane refractive index that was determined by linearly polarized laser light parallel to the film plane; *n*_TM_ was the out-of-plane refractive index, which was determined by linearly polarized laser light perpendicular to the film plane.

The thermal stability of the PI films was measured by thermogravimetric analysis (TGA) on a TA-Q50 thermal analysis system (TA Instruments, New Castle, DE, USA) at a heating rate of 20 °C/min under nitrogen ranged from 40 °C to 760 °C. Differential scanning calorimetry (DSC) was conducted on the PI films using a TA-Q100 thermal analysis system in a standard aluminum crucible at a heating rate of 10 °C/min in nitrogen. The dimensional thermal stability of the PI films was tested with a thermomechanical analyzer (TMA) using a TA-Q400 thermal analysis system (TA Instruments, New Castle, DE, USA) in nitrogen at a heating rate of 10 °C /min. The coefficients of linear thermal expansion (CTE) of the PI films in the range of 50–200 °C were recorded.

The dielectric properties of the PI composite films were measured using an Agilent 4294A precise impedance analyzer (Palo Alto, CA, USA) at room temperature. The test frequency range is 10^3^–10^6^ Hz. The dielectric constant of PI films was calculated by the following Equation (2) [25]:(2)$$D_{k}=\frac{Cd}{D_{0}A}$$
where, *D*_k_ is the dielectric constant; *C* is the capacitance (F); *d* is the thickness of the sample (m); *D*_0_ is the dielectric constant of vacuum, 8.854 × 10^−12^ F/m; *A* is the area of the sample (m^2^).

The atomic oxygen (AO) erosion performance of PI films was obtained in a ground-based AO effects simulation facility in the Beijing Institute of Spacecraft Environment Engineering [12]. The size of the PI film sample was 20 × 20 × 0.05 mm^3^. The total AO fluence for this test was 4.02 × 10^20^ atoms/cm^2^, and the mass loss of PI films after AO erosion was recorded. The erosion yield (*E*_s_) of the PI sample is calculated by the following Equation (3) [26]:(3)$$E_{s}=\frac{\Delta M_{s}}{A_{s}\rho_{s}F}$$
where, *E*_s_ = erosion yield of PI sample (cm^3^/atom); Δ*M*_s_ = mass loss of PI sample (g); *A*_s_ = surface area of the PI sample exposed to AO (cm^2^); *ρ*_s_ = density of the sample (g/cm^3^); *F* = AO fluence (atoms/cm^2^). *A*s Kapton^®^ film has a specific erosion yield of 3.0 × 10^−24^ cm^3^/atom [27], and all the present PI samples in this test are supposed to possess similar density and exposure area with Kapton^®^ film, therefore, the simplified Equation (4) could be used to calculate the *E*_s_ of the PIs:(4)$$E_{s}=\frac{\Delta M_{s}}{\Delta M_{\mathrm{Kapton}}}E_{\mathrm{Kapton}}$$
where, *E*_Kapton_ was the erosion yield of Kapton^®^, 3.0 × 10^−24^ cm^3^/atom; Δ*M*_Kapton_ was the mass loss of Kapton^®^.

### 2.3. Synthesis of Poly(amic acid)s and Preparation of PI Composite Films

A series of fluoro-containing PI films combined with different loading content of TSP-POSS additives (0, 5 wt%, 10 wt%, 15 wt%, 20 wt%, and 25 wt% based on the total solid films) were designed and prepared. For this target, the poly(amic acid) (PAA) precursors with the TSP-POSS fillers were prepared first. Then, the PAA/TSP-POSS precursors were thermally dehydrated to finish the imidization reaction to afford the final PI/TSP-POSS composite films. The derived PAA solutions were named as “6FPAA-X”, where “6FPAA” stands for the fluoro-containing PAA and “X” represents the weight percent content of the TSP-POSS additives. The afforded PI composites films were named as “6FPI-X”, where “6FPI” stands for the fluoro-containing PI matrix and “X” stands for the weight percent content of the TSP-POSS additives in the composite films. The detailed synthesis procedure could be explained by the preparation procedure of 6FPI-5. First, BDAF (51.8450 g, 100 mmol) and TSP–POSS (5.0668 g, 5.44 mmol) were added into a 1000-mL four-necked flask equipped with a cold-water bath, an anchor-type mechanical stirrer (Shanghai Shupei Laboratory Equipment Co., Ltd, Shanghai, China), and a nitrogen inlet and outlet. Then, the ultra-dry DMAc solvent (400.0 g) was added to the flask. The mixture was stirred at 10 °C for 15 min under a nitrogen flow (20 mL/min) to afford a clear solution. Subsequently, 6FDA (44.4240 g, 100 mmol) was added together with an additional DMAc (60.0 g). The solid content of the polymerization system was 18 wt%. After 3 h, the cold-water bath was removed. The reaction mixture was stirred at room temperature for another 20 h. Then, the afforded viscous and clear 6FPAA-5 solution was diluted to be 15 wt%. The obtained solution was filtered through a 0.50-µm polytetrafluoroethylene (PTFE) filter to remove any undissolved impurities.

The other 6FPAA-X solutions were synthesized on the basis of the procedures mentioned above. The detailed formulas for the preparation of the polymers were listed in Table 1.

Secondly, the 6FPI-5 film was prepared by using the 6FPAA-5 as the starting materials. The purified and de-foamed 6FPAA-5 solution was spin-coated on a quartz wafer with a diameter of 10.16 cm (4 inches). The thicknesses of the PI films from 10 μm to 100 μm were adjusted by the spin-coating parameters. The quartz wafer with the coated 6FPAA-5 solution were placed in a clean oven circulated with dry nitrogen and then thermally baked with the conditions of 80 °C/1 h, 120 °C/1 h, 150 °C/1 h, 180 °C/1 h, 250 °C/1 h, and 300 °C/1 h, respectively. Then, the quartz wafer was immersed into the deionized water to afford the free-standing 6FPI-5 film.

The other 6FPI-X films, including 6FPI-0, 6FPI-10, 6FPI-15, 6FPI-20, and 6FPI-25 were prepared on the basis of the similar procedures mentioned above.

## 3. Results and Discussion

### 3.1. FPI-X Composite Films Preparation

In order to develop high-performance polymer substrates with both excellent AO resistance and low-*D*_k_ values for potential applications as antenna components for LEO spacecraft, a series of PI nanocomposite polymers were synthesized according to the synthesis routes shown in Figure 2. The PI matrix was based on the fluoro-containing PI (6FDA-BDAF) system, which has been well known as aerospace-class PI materials servicing in the geosynchronous earth orbit (GEO) spacecraft due to the excellent combined properties, including excellent ultraviolet-light resistance, good thermal stability, high optical transparency, and good mechanical and dielectric properties [28]. Especially, the PI (6FDA-BDAF) matrix has been proven to exhibit low-*D*_k_ and low-*D*_f_ features in the wide frequency range [29]. Thus, the system might be one of the best candidates for antenna substrate applications. However, the PI system was highly sensitive to AO exposure and might lose the original valuable properties in LEO environments. The introduction of TSP-POSS into the matrix might be an effective method to enhance the AO-resistant properties of the pristine polymer. TSP-POSS has been widely used for the improvements of AO resistance of common Kapton^®^ film [30]. The polar trisilanol groups in TSP-POSS could efficiently increase the miscibility of the compounds with the polar poly(amic acid) (PAA) matrix. This made the TSP-POSS a good candidate as additives for developing high-performance nanocomposites in which the nano-level uniform dispersion and optical transparency were especially concerned. In the current research, both the PI (6FDA-BDAF) matrix and the TSP-POSS additives are thought to be good candidates for developing antenna substrates with potential applications in LEO.

The TSP-POSS additives exhibited good miscibility with both the PAA and PI matrix, which could be found in Figure 3, in which the appearances of the 6FPAA-X PAA solutions (Figure 3a) and the typical 6FPI-25 film (Figure 3b) were presented. It can be visually observed that the PI composite films with the TSP-POSS loading amounts as high as 25 wt% could still maintain good optical transparency. This is mainly attributed to the nano-level miscibility between the matrix and the additives. No obvious particle accumulation and phase separation occurred between the continuous phase (PI) and the disperse phase (TSP-POSS).

The chemical structures of the PI films were confirmed by the ATR-FTIR measurements, as shown in Figure 4. All the polymer systems exhibited similar absorption behaviors in the spectra due to the similar structural compositions. For example, the characteristic absorptions of imide groups, including the asymmetrical and symmetrical stretching vibration of C=O in imides at about 1786 cm^−1^ and 1721 cm^−1^, respectively, and stretching vibration of C-N in imides at about 1375 cm^−1^ were all clearly detected for all of the samples. The differences among the spectra mainly came from the characteristic absorptions of Si-O-Si structure in TSP-POSS additives at 1132 cm^−1^. It was not observed in the spectrum of 6FPI-0 without TSP-POSS filler. However, it was clearly found in the spectra of the composite films. Meanwhile, with the increase of TSP-POSS additives, the Si-O-Si absorption peak strength gradually increased, indicating that POSS fillers were successfully added into the polymers.

The XRD plots of the PI films together with the TSP-POSS additive were shown in Figure 5. TSP-POSS exhibited obvious crystalline features with scattering angles in the range of 10° to 30°. However, only blunt absorption peaks instead of sharp ones for TSP-POSS were observed in the spectra of the PI composite films, indicating the good compatibility between TSP-POSS and PI matrix.

The XPS spectra of the PI films shown in Figure 6 also proved the successful incorporation of the TSP-POSS additives into the PI matrix. This could be confirmed from the intensity of the Si2p absorptions in the spectra. For the pristine 6FPI-0, no Si2p absorptions were observed. However, they could be detected in the spectra of the PI composite films (6FPI-5. 6FPI-10, 6FPI-15, 6FPI-20, and 6FPI-25). Meanwhile, with the amounts of the TSP-POSS in the composite films increased, the strength of the Si2p absorption peaks gradually increased. The characteristic absorptions of C1s, N1s, O1s, and F1s were detected for all of the PI samples.

The surface morphology of the PI films was measured by AFM measurements and the obtained images were shown in Figure 7, together with the average roughness (*R*_a_) and the root mean square roughness (*R*_q_) of the PI films. It can be observed that the pristine 6FPI-0 film showed low roughness with the *R*_a_ and *R*_q_ values of 3.81 nm and 5.09 nm, respectively. The incorporation of TSP-POSS additives slightly increased the roughness of the PI films. For example, the *R*_a_ values of the composite films increased from the original 3.81 nm for 6FPI-0 to 13.9 nm for 6FPI-5. Then, the *R*_a_ values of the composite films decreased to the level below 10 nm with the increase of the contents of TSP-POSS additives. This changing trend indicated that the distribution of TSP-POSS additives in the PI composite films tended to be uniform when the loading amounts reached over 5 wt% in the composite films.

All the characterization results mentioned above proved the uniform dispersion of the TSP-POSS additives into the PI matrix. The original optical transparency of the pristine 6FPI-0 was visually maintained. The influence of the TSP-POSS additives on the other properties of the derived PI composite films will be further studied.

### 3.2. Optical and Dielectric Properties

The influence of the incorporation of the TSP-POSS additives on the optical properties of the PI composite films was quantitatively investigated by UV-Vis, refractive indices, and CIE Lab optical parameters measurements, respectively. The obtained optical data are summarized in Table 2. The UV-Vis spectra of the PI films are shown in Figure 8. It can be seen that the incorporation of TSP-POSS apparently increased the optical transparency of the composite films. For example, the pristine 6FPI-0 film exhibited the cutoff wavelength (*λ*_cut_) of 331 nm and the transmittance at the wavelength of 450 nm (*T*_450_) of 82.8% at the thickness of 25 μm, respectively. However, for the PI composite films, these optical parameters became better with the incorporation of the TSP-POSS fillers. 6FPI-25 showed a *λ*_cut_ value of 292 nm, which was 39 nm lower than that of the 6FPI-0 film. The *T*_450_ value of the 6FPI-25 film (84.7%) was also higher than the 6FPI-0 film. This phenomenon might be due to the excellent dispersion of the TSP-POSS additive into the PI matrix. The bulky and cage-like molecular structures of TSP-POSS fillers apparently decreased the molecular chain packing densities in the PI composite films, thus facilitating the penetration of visible light. It is worthy of notice that even the high loading of TSP-POSS up to 25 wt% did not deteriorate the optical transmittance of the PI composite films. This is quite beneficial for developing high-performance inorganic/organic hybrid composite films with tailored optical properties.

The refractive indices (*n*_AV_) of the PI films were measured with the prism coupler and the data were tabulated in Table 2. The influence of TSP-POSS additives on the optical properties of the PI composite films could also be deduced from the changing trends of the *n*_AV_ values of the films. Although 6FPI-0 and 6FPI-25 films showed very close *n*_AV_ values, the *n*_AV_ values of the PI composite films first increased with the increasing loading of TSP-POSS from 0 to 10 wt% in the composite films. Then, the *n*_AV_ values of the PI composite films began to decrease when the loading of TSP-POSS was higher than 10 wt%. The –OH groups in TSP-POSS possessed high molar refraction, while the cage-type silsesquioxane groups showed high molar volume. Therefore, when the TSP-POSS content was low (<10 wt%), the refractive indices of the composite films increased slightly with the presence of the –OH structure; however, when the TSP-POSS content was high, the cage-type silsesquioxane groups gradually played a predominant role and reduced the refractive indices of the composite films.

The three-dimension (3D) plots of CIE Lab parameters of the PI films were shown in Figure 9, and the corresponding data are summarized in Table 2. The pristine 6FPI-0 film showed low color parameters with the lightness (*L**) of 95.78, the yellow indices (*b**) of 4.40, and the haze of 2.05%. With the increase of the contents of TSP-POSS additives, the *L** values of the films decreased first and then increased, while the *b** values of the films showed a trend of increasing first and then decreasing. The –OH groups in TSP-POSS were sensitive to the high-temperature oxidative environments during the fabrication of PI films from 80 to 300 °C. Therefore, when the TSP-POSS additives were incorporated into the PI matrix at low contents, the optical parameters of the composite films slightly deteriorated. However, when the contents of TSP-POSS additives reached a high level (>15 wt%), a partly continuous TSP-POSS phase gradually formed in the composite films. This is beneficial for improving the lightness and decreasing the yellow indices of the composite films. As for the haze values of the PI films, they increased with the gradual increase of the contents of TSP-POSS additives. 6FPI-25 exhibited the highest haze value of 5.14% in the samples, which was apparently higher than that of the pristine 6FPI-0 (haze = 2.05%). This suggested that although very uniform dispersion was formed between the PI matrix and the TSP-POSS additives, microscopic aggregation might occur for the TSP-POSS fillers, especially at a higher content.

Although the incorporation of TSP-POSS showed some positive and negative effects on the optical properties of the PI composite films, the optical properties of the polymers were not the main concerns in the current research. Figure 10 shows the curve of the dielectric constants of the PI films as a function of frequency. All the PI films showed stable *D*_k_ values in the frequency range from 10^3^ to 10^6^ Hz. The 6FPI-0 film, an intrinsically low-*D*_k_ PI [28], showed the *D*_k_ value of 2.84 at 1 MHz and the *D*_f_ value of 0.009. The addition of TSP-POSS additives apparently reduced the *D*_k_ values of PI composite films, while the *D*_f_ values maintained the same level. With the increase of the TSP-POSS contents, the *D*_k_ values of the PI films decreased from 2.84 (1 MHz) of 6FPI-0 to 2.52 (1 MHz) of 6FPI-25, while the *D*_f_ values correspondingly changed from 0.009 (1 MHz) to 0.011 (1 MHz). According to the Clausius–Mossotti equation, the increase of the internal space of materials can reduce the dielectric constant [31]. Herein, the introduction of TSP-POSS additives with a large bulky structure can break the tight packing of PI molecular chains, thus reducing the dielectric constant of PI composite films. The low-*D*_k_ feature of the current PI composite films is mainly attributed to the synergistic effects of the bulky hexafluoroisopropylene groups in PI matrix and the bulky silsesquioxane groups in TSP-POSS additives. The low *D*_k_ value of 2.52 at 1 MHz makes the 6FPI-25 a good candidate as high-performance substrates for spacecraft antenna.

### 3.3. Thermal Properties

The influence of the incorporation of TSP-POSS on the thermal stability of the PI composite films was studied by TGA, DSC, and TMA measurements, respectively, and the data are tabulated in Table 3. The TGA curves of the PI films measured in nitrogen environments were shown in Figure 11. All the PI films maintained their initial weights when the temperature was over 450 °C and the *T*_5%_ values were recorded in the range of 494–520 °C. The pure TSP-POSS additives exhibited good thermal stability with the *T*_5%_ values over 360 °C in nitrogen [32], which endowed the PI composite films with good thermal stability. The PI films left 45.3~53.8% of their original weights at 760 °C. The *R*_w760_ values of the PI films decreased with the order of 6FPI-0 (53.8%) > 6FPI-5 (53.6%) > 6FPI-10 (48.4%) > 6FPI-15 (48.1%) > 6FPI-20 (47.4%) > 6FPI-25 (45.3%). This trend is quite different from the thermal decomposition behaviors of conventional POSS/polymer composite films, which is peculiar to the POSS combined fluoro-containing systems. This phenomenon is mainly attributed to the releasing of gaseous fluorine–silicon compounds derived from the reaction of decomposed fluoro species and the silicon elements at elevated temperatures [12].

The DSC diagram of PI films was shown in Figure 12. It can be seen from the figure that the incorporation of TSP-POSS has little effect on the *T*_g_ values of PI films. The *T*_g_ value of the pristine 6FPI-0 film was very close to those of the PI composite films. On one hand, the TSP-POSS possessed bulky molecular chains, which were not easy to move at elevated temperatures. On the other hand, the TSP-POSS additives might not have a strong interaction with the PI matrix due to the absence of active groups in the PI matrix. These factors lead to that TSP-POSS additives have little effect on the glass transition behaviors of the PI composite films.

Although the incorporation of TSP-POSS additives into the PI matrix did not bring negative effects on the *T*_g_ values of the PI composite films, they deteriorated the high-temperature dimensional stability of the polymers, as could be evidenced by the TMA plots of the PI composite films in Figure 13. The pristine 6FPI-0 film itself possessed the CTE value of 62.3 × 10^−6^/K due to the presence of flexible hexafluoroisopropylene and ether linkages in the polymer. When the TSP-POSS additives were introduced, the composite films showed further increased CTE values at elevated temperatures. For example, 6FPI-25 exhibited a CTE value of 72.4 × 10^−6^/K, which was obviously higher than that of the pristine 6FPI-0 film. Apparently, the plasticization effects of the TSP-POSS fillers increased the CTE values of the composite films. The deteriorated high-temperature dimensional stability might be disadvantageous for their applications as substrates for spacecraft antennas. Modification of the high-temperature dimensional stability of the current PI composite films will be investigated in our future work.

### 3.4. AO Erosion Properties

The AO erosion behaviors of the PI composite films were evaluated in a ground simulation facility. The actual total AO exposure fluence was 4.02 × 10^20^ atoms/cm^2^, which is equivalent to nearly half a year of AO exposure in LEO at the altitude of 500 km [33]. Kapton^®^ films were used as a reference, whose erosion yield (*E*_S_) of 3.0 × 10^−2^^4^ cm^3^/atom was taken as a standard. The PI films after AO exposure were named as “6FPI-X-AO”, where X stands for the weight percent of TSP-POSS additives in the composite films. Figure 14 shows the *E*_S_ values of the PI films, together with the appearances of the typical 6FPI-0-AO and 6FPI-25-AO. The experimental data for the AO erosion evaluation are listed in Table 4. It could be clearly seen that the *E*_S_ values of the PI films apparently decreased with the increase of the TSP-POSS contents in the PI composite films. 6FPI-25 sample with the highest TSP-POSS contents showed the lowest *E*_S_ value of 0.40 × 10^−2^^4^ cm^3^/atom, which was nearly an order of magnitude lower than that of Kapton^®^, and also nearly an order of magnitude lower than that of the pristine 6FPI-0 film (*E*_S_ = 2.60 × 10^−2^^4^ cm^3^/atom). Thus, the incorporation of TSP-POSS additives could efficiently enhance the AO-resistant properties of the PI composite films.

It could also be observed from Figure 14 that the PI films became opaque after AO exposure. Figure 15 presents the UV-Vis spectra of the AO-exposed 6FPI-X-AO films after the AO exposure with the dose of 4.02 × 10^20^ atoms/cm^2^ and the optical transmittance at 450 nm of the AO-exposed PI films (*T*_450AO_) are shown in Table 2. First, when comparing the *T*_450_ data of the PI films shown in Figure 8 and Figure 15, an apparent decrease of the optical transparency for the PI films before and after AO exposure could be clearly observed. For example, the 6FPI-25 film showed the *T*_450_ value of 84.7%, while the 6FPI-25-AO film showed a value of 37.1%. This indicated that AO exposure severely deteriorates the optical properties of PI films. Secondly, the *T*_450AO_ values of 6FPI-X-AO films decreased with the order of 6FPI-0-AO > 6FPI-25-AO > 6FPI-20-AO > 6FPI-15-AO > 6FPI-10-AO > 6FPI-5-AO, which was different from that of the 6FPI-X films. Thus, for the PI composite films, the influence of the AO exposure on the deterioration of the optical transparency might be influenced by many factors, which will be discussed below.

It has been well established that some specific elements, such as silicon, phosphorus, titanium, and so on could endow the PI films self-passivating or self-healing features in AO environments due to the potential ability for forming the passivation layers of silicate, phosphate, titanate, or the other inorganic oxides when the functional PI films were exposed to AO environments [28]. For the current PI composite films, such passivation layers were also detected, which could be found in the SEM measurement shown in Figure 16. For the pristine 6FPI-0-AO film, carpet-like micromorphology was observed after AO exposure, which is the typical characteristic of the AO-eroded polymer surface. Linear weight loss was found for the system. With the contents of the TSP-POSS additives in PI films increased from 0 to 15 wt%, the AO-eroded surface gradually became rough. When the content of TSP-POSS additives was over 20 wt%, the AO-eroded surface gradually became smooth. For the 6FPI-25-AO sample, clear passivation layers were observed on the surface. For the 6FPI-X-AO films, the rough surface might prohibit the transmission of visible light, while the smooth surface was beneficial for the penetration of visible light. Thus, the 6FPI-X-AO films showed the *T*_450_ trends showing in Figure 15.

The surface roughness of AO-exposed PI films was investigated by AFM measurements and the change of surface morphology was also confirmed. The 2D AFM images of AO-exposed PI films were shown in Figure 17. It can be observed that the surface roughness of the AO-exposed 6FPI-X-AO films was markedly increased than those of unexposed 6FPI-X-AO films, which are shown in Figure 7. The roughness of the AO-exposed PI films showed a downward trend when TSP-POSS content was higher than 15 wt%. The 6FPI-25-AO composite film exhibited the lowest roughness with the *R*_a_ and *R*_q_ values of 55.6 nm and 70.4 nm, respectively. It might be due to the formation of the continuous and dense passivation layer onto the surface of AO-exposed 6FPI films.

The relative atomic concentrations of the AO-exposed and unexposed films measured by XPS were tabulated in Table 5. As shown in the table, the relative atomic concentration of C element onto the surface of the 6FPI-X-AO films (except for 6FPI-0-AO) decreased apparently after AO exposure, whereas the relative atomic concentration of O and Si elements increased greatly. The change of Si element on the surface of PI composite films before and after AO exposure was further analyzed, and the obtained high-resolution XPS spectra were shown in Figure 18. It can be clearly observed that the peak intensity of Si2p on the surface of AO-exposed films gradually increased with the increasing contents of TSP-POSS additives, and the binding energy of Si2p also increased. For example, the binding energy peak of Si2p on the surface of the 6FPI-15 film shifted from 102.31 eV before exposure to 103.34 eV after exposure. The apparent shift of the binding energy peak of Si2p at ~1.0 eV after AO exposure indicated that the Si element changes from Si-O-Si structure on the surface of the 6FPI-15 film to SiO_2_ structure for 6FPI-15-AO film. The molar ratio of Si element to O element is close to 1:2 as shown in Table 5. These results demonstrated that the formation of the silica passivation layer on the surface of PI composite films, which could be a protective layer for the underlying polymer materials.

## 4. Conclusions

In summary, an optically transparent film with good dielectric properties and good AO resistance in simulated LEO environments has been successfully prepared. The properties of 6FPI were endowed with the synergistic effects of fluoro-containing PI matrix and the TSP-POSS additives. The 6FPI-25 composite film at the thickness of 25 μm possessed high optical transparency (*λ*_cut_ = 292 nm, *T*_450_ = 84.7%, haze = 5.14%) and was nearly colorless (*a** = −1.34, *b** = 3.80, *L** = 95.78). Meanwhile, the 6FPI-25 film possessed good dielectric properties with low *D*_k_ and *D*_f_ (*D*_k_ = 2.52 at 1 MHz, *D*_f_ = 0.010 at 1 MHz) and good thermal stability with the *T*_5%_ of 494 °C. Moreover, the AO erosion yield of 6FPI-25 composite film was 0.40 × 10^−24^ cm^3^/atom after AO exposure at a fluence of 4.02 × 10^20^ atoms /cm^2^. The XPS and SEM measurements confirmed that the passivation layer formed on the surface of the exposed film could protect the underlying material from further AO attack. The transparent PI composite films with good dielectric properties and long-term sustainability in AO environments are good candidates for antenna substrates in LEO aerospace applications.

## Figures and Tables

**Figure 1 nanomaterials-11-01886-f001:**
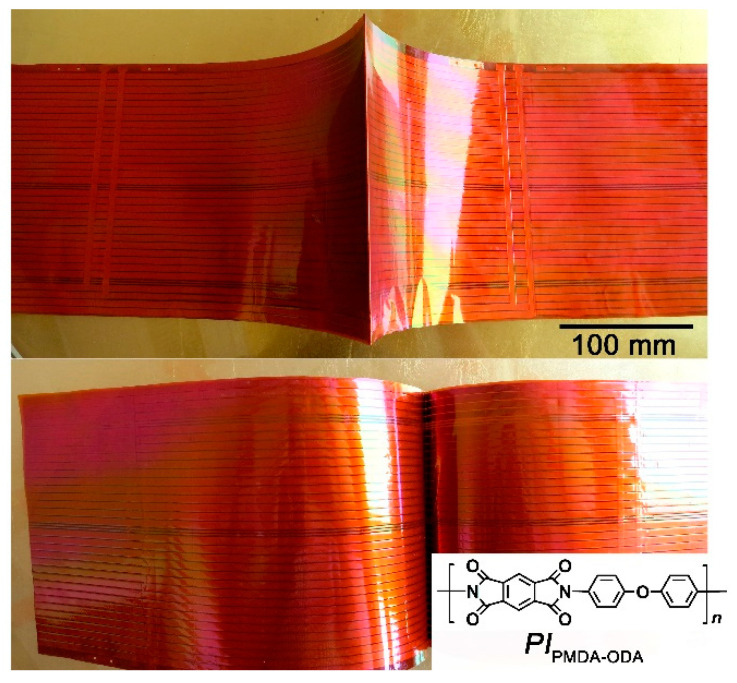
LEO spacecraft antennas based on PI_PMDA-ODA_ substrates.

**Figure 2 nanomaterials-11-01886-f002:**
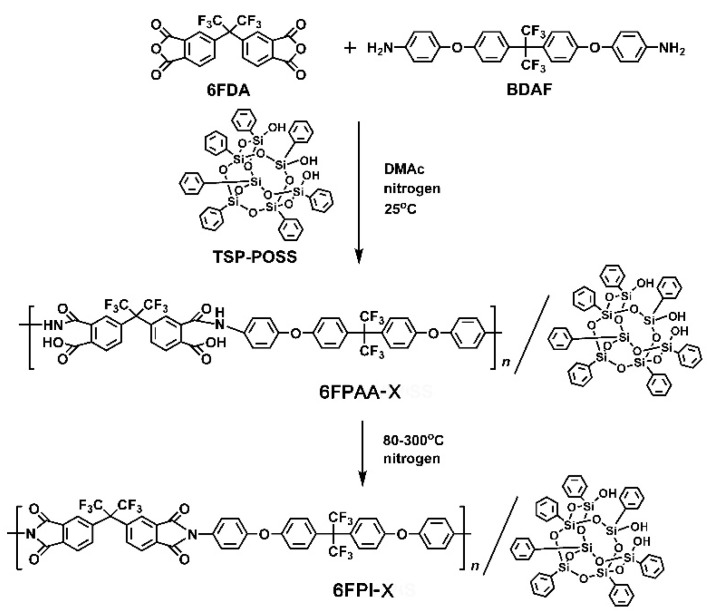
Synthesis of 6FPAA-X solutions and preparation of 6FPI-X composite films.

**Figure 3 nanomaterials-11-01886-f003:**
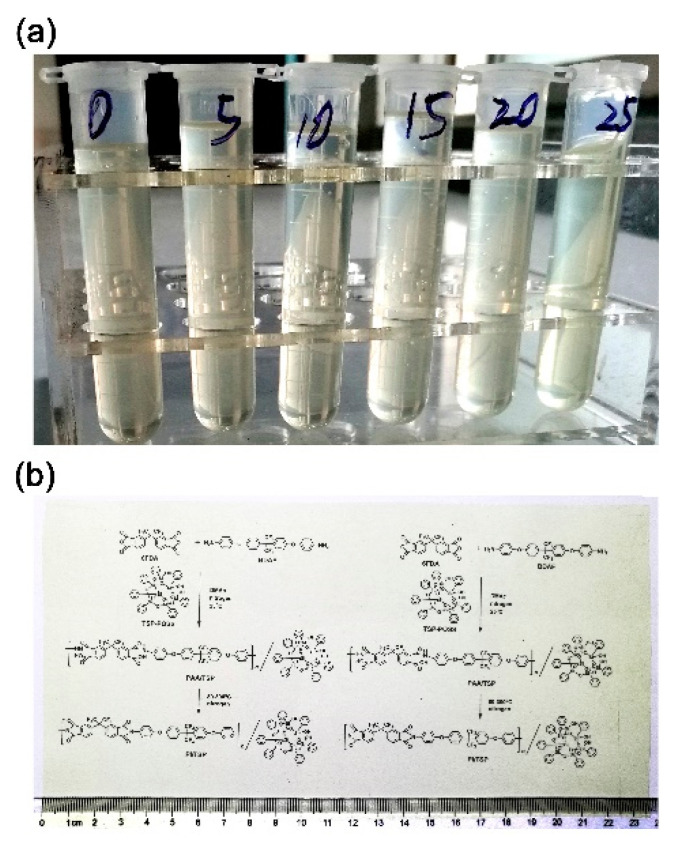
Appearance of polymers. (**a**) 6FPAA-X solutions (from left to right: 6FPAA-0, 5, 10, 15, 20, and 25); (**b**) 6FPI-25 composite film.

**Figure 4 nanomaterials-11-01886-f004:**
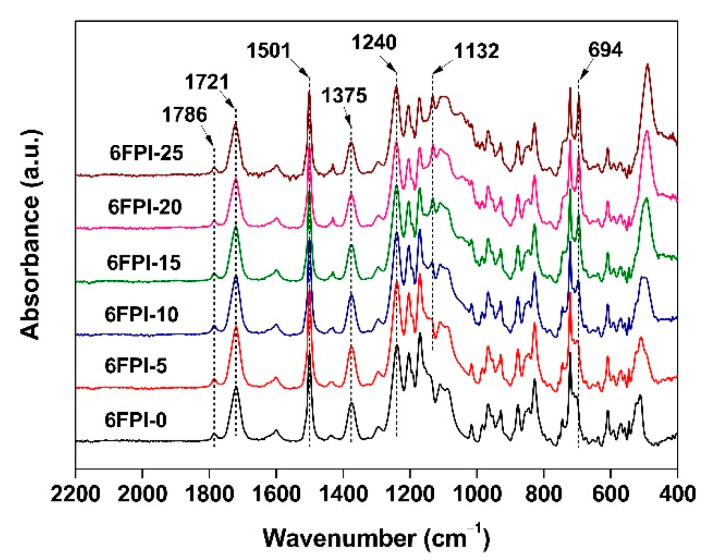
ATR-FTIR spectra of 6FPI-POSS composite films.

**Figure 5 nanomaterials-11-01886-f005:**
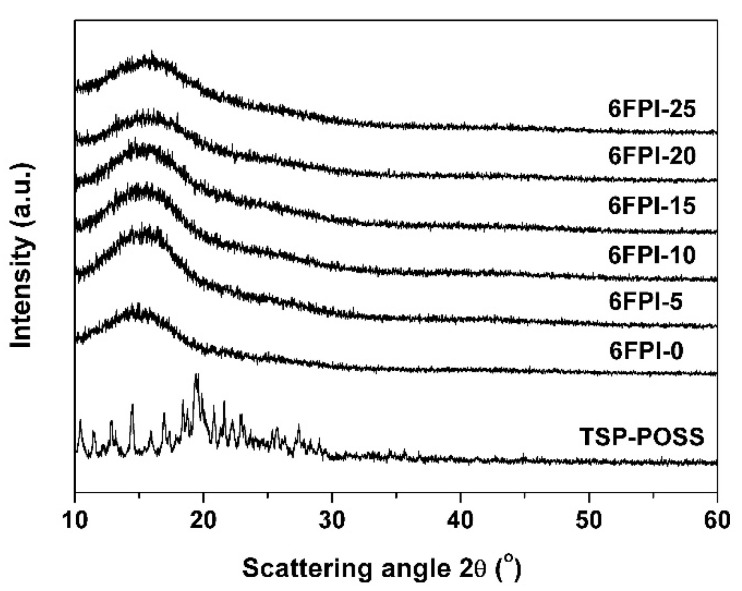
XRD plots of 6FPI-POSS composite films.

**Figure 6 nanomaterials-11-01886-f006:**
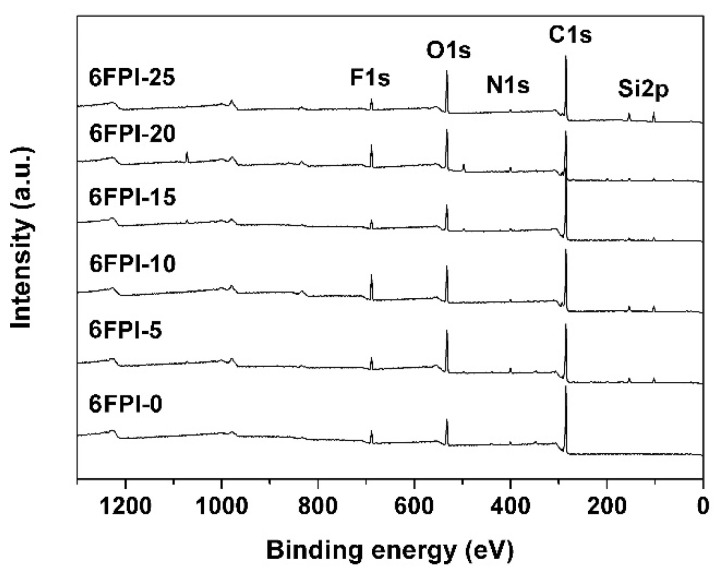
XPS spectra of 6FPI-POSS composite films.

**Figure 7 nanomaterials-11-01886-f007:**
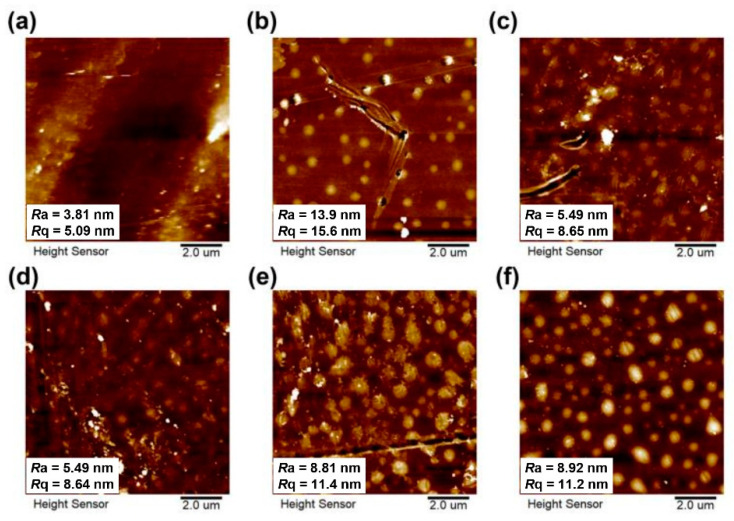
AFM images of 6FPI-POSS composite films with the surface roughness values. (**a**) 6FPI-0; (**b**) 6FPI-5; (**c**) 6FPI-10; (**d**) 6FPI-15; (**e**) 6FPI-20; (**f**) 6FPI-25.

**Figure 8 nanomaterials-11-01886-f008:**
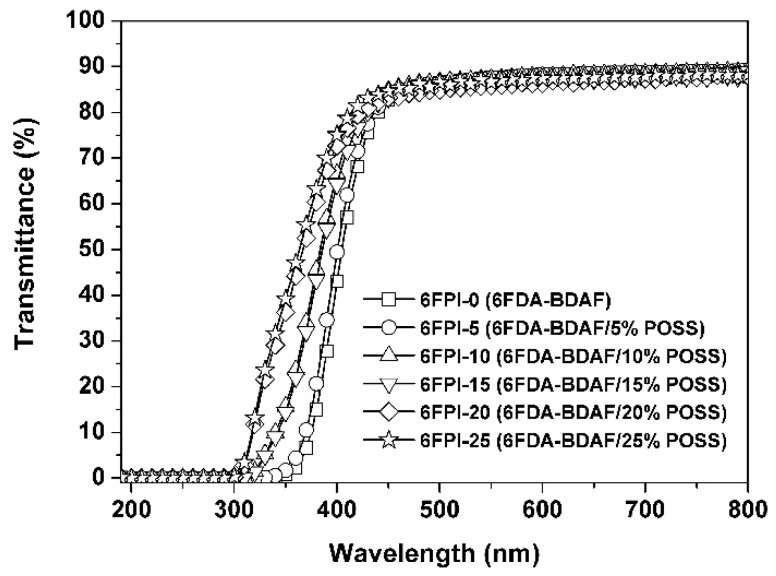
UV-Vis spectra of 6FPI-POSS composite films.

**Figure 9 nanomaterials-11-01886-f009:**
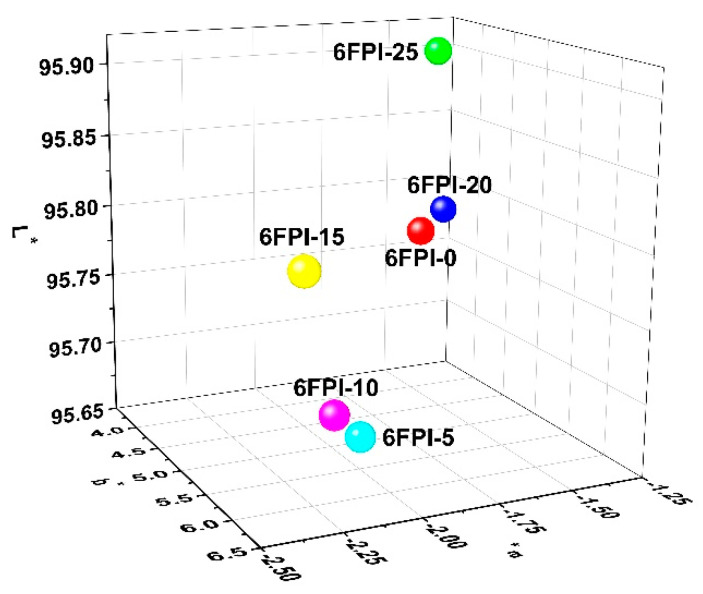
CIE Lab parameters of 6FPI-POSS composite films.

**Figure 10 nanomaterials-11-01886-f010:**
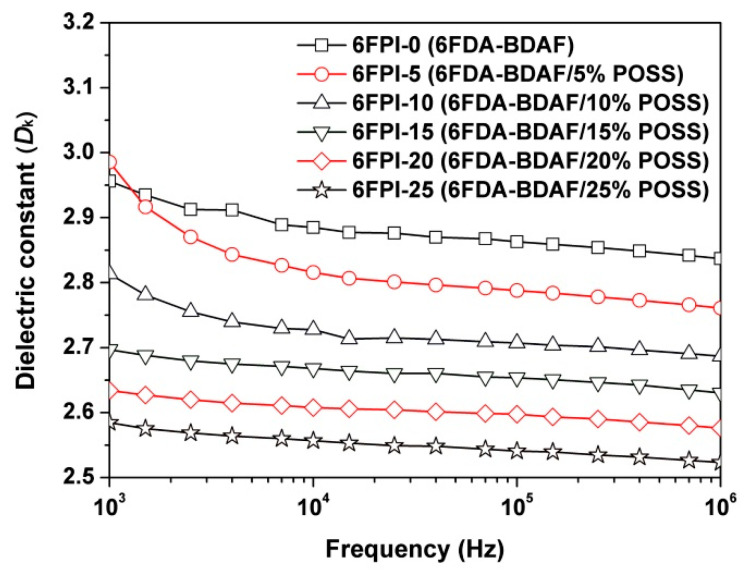
Dielectric parameters of 6FPI-POSS composite films.

**Figure 11 nanomaterials-11-01886-f011:**
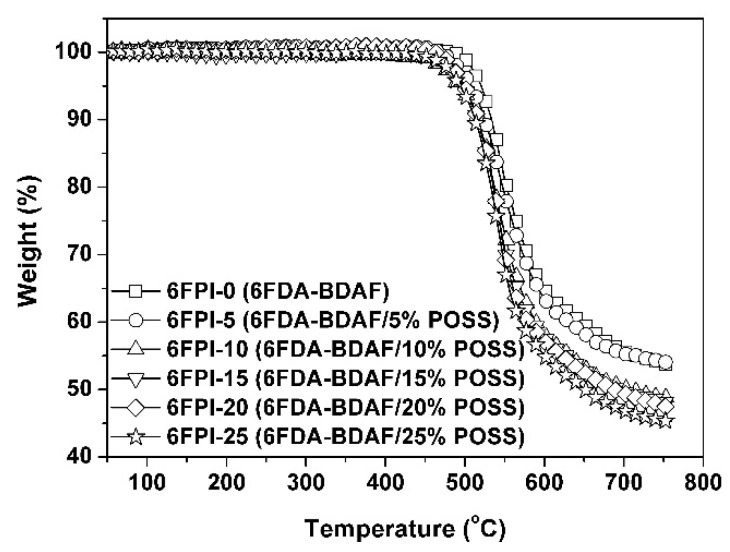
TGA curves 6FPI/POSS composite films under nitrogen flow.

**Figure 12 nanomaterials-11-01886-f012:**
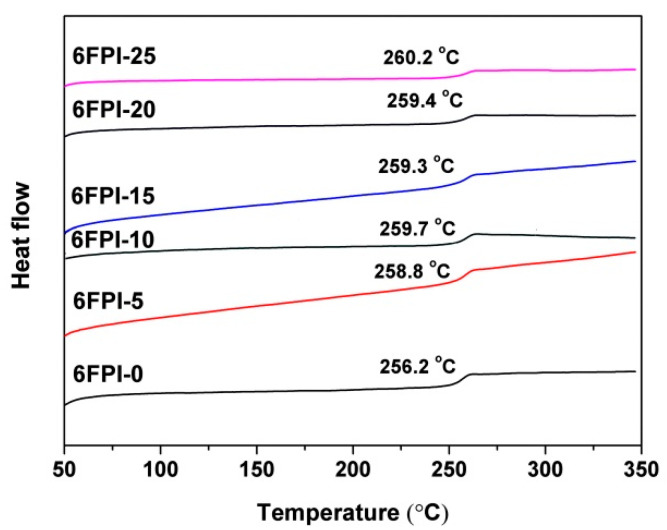
DSC curves 6FPI/POSS composite films under nitrogen flow.

**Figure 13 nanomaterials-11-01886-f013:**
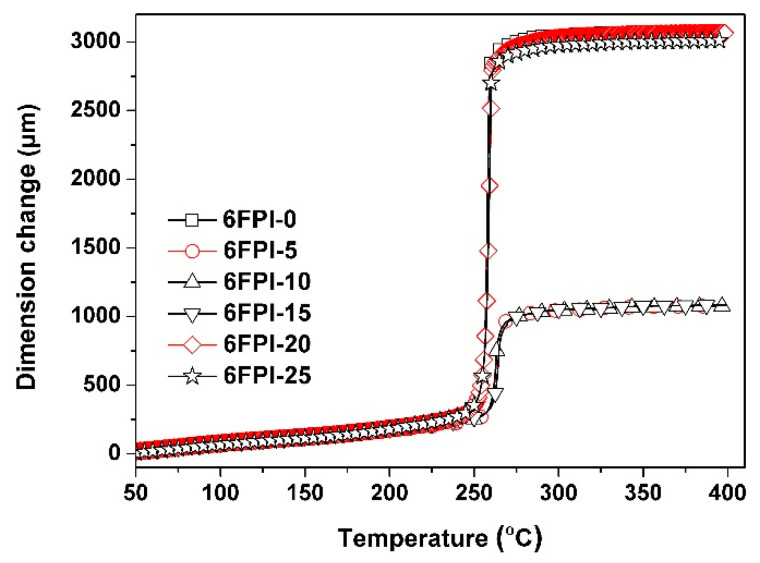
TMA curves 6FPI/POSS composite films at a heating rate of 10 °C/min under nitrogen flow.

**Figure 14 nanomaterials-11-01886-f014:**
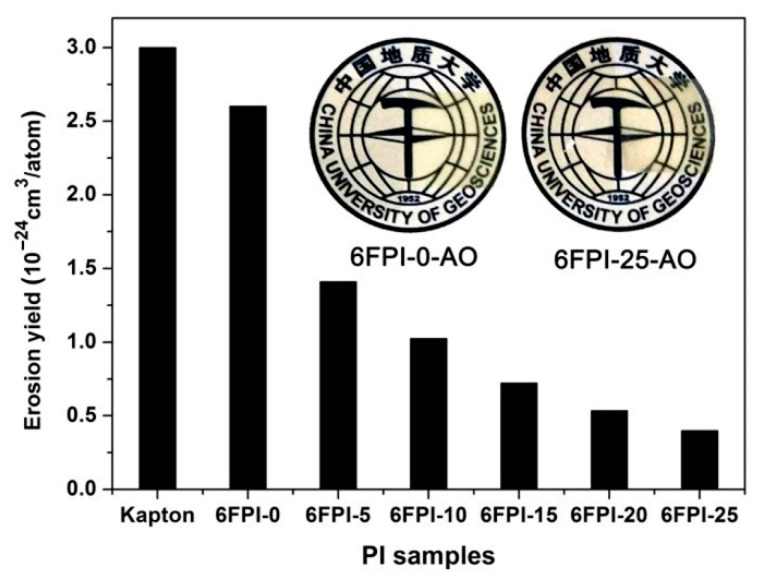
AO erosion yields of the 6FPI-POSS composite films (Insert: Appearance of representative 6FPI-0-AO and 6FPI-25-AO films).

**Figure 15 nanomaterials-11-01886-f015:**
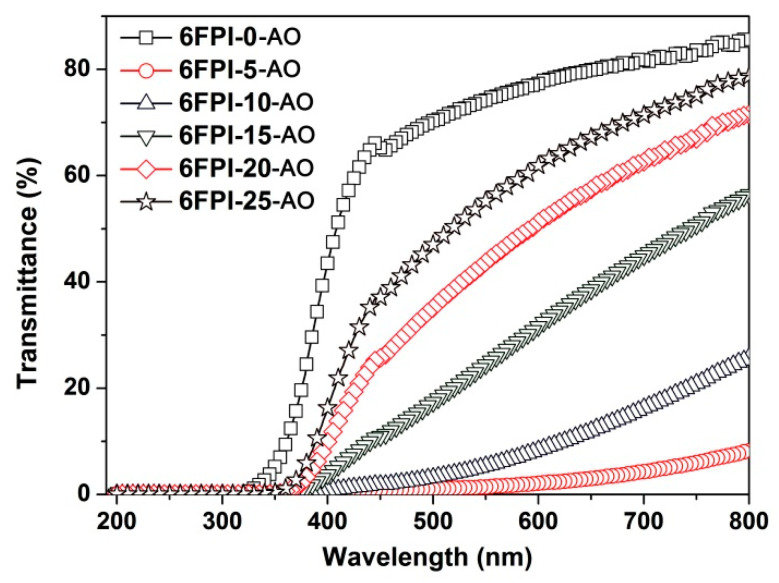
UV-Vis spectra of 6FPI-POSS films after AO exposure.

**Figure 16 nanomaterials-11-01886-f016:**
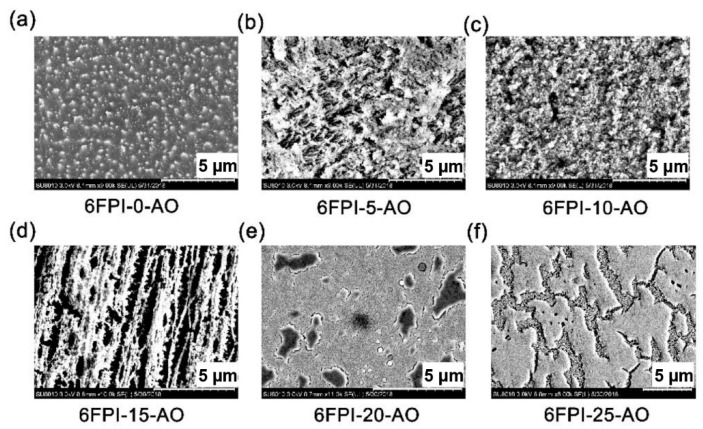
SEM images of 6FPI-X-AO films after AO exposure. (**a**) 6FPI-0-AO, (**b**) 6FPI-5-AO, (**c**) 6FPI-10-AO, (**d**) 6FPI-15-AO, (**e**) 6FPI-20-AO, (**f**) 6FPI-25-AO.

**Figure 17 nanomaterials-11-01886-f017:**
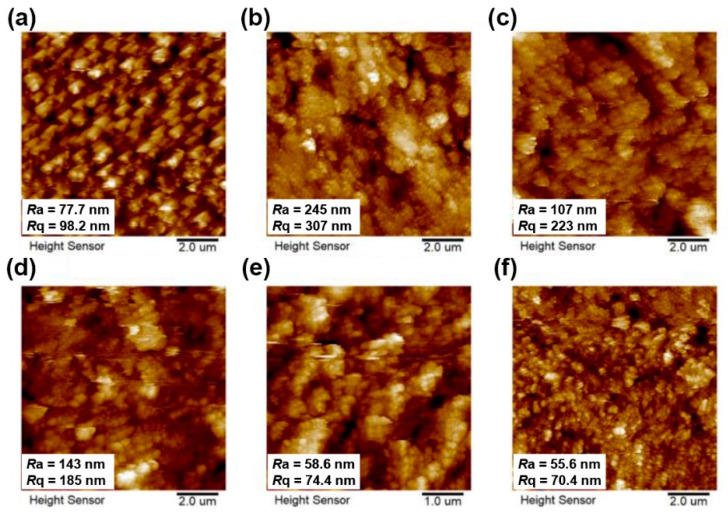
The 2D AFM images (10.0 µm × 10.0 µm) of 6FPI-X-AO films after AO exposure. (**a**) 6FPI-0-AO, (**b**) 6FPI-5-AO, (**c**) 6FPI-10-AO, (**d**) 6FPI-15-AO, (**e**) 6FPI-20-AO, (**f**) 6FPI-25-AO.

**Figure 18 nanomaterials-11-01886-f018:**
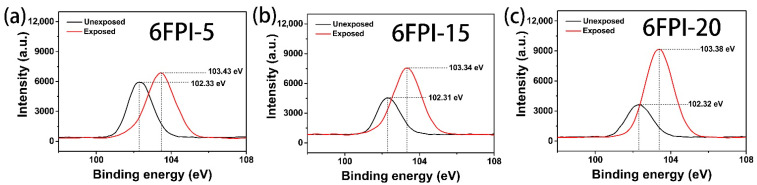
XPS spectra of Si2p for AO-exposed and unexposed 6FPI-POSS composite films with total AO fluence of 4.02 × 10^20^ atom/cm^2^. (**a**) 6FPI-5; (**b**) 6FPI-15; (**c**) 6FPI-20.

**Table 1 nanomaterials-11-01886-t001:** Formulas for the synthesis of 6FPAA-X solutions.

PAA	6FDA (*M* = 444.24 g/mol)	BDAF (*M* = 518.45 g/mol)	TSP-POSS (*M* = 931.34 g/mol)	DMAc (g)
6FPAA-0	44.4240 g (100 mmol)	51.8450 g (100 mmol)	0	439.0
6FPAA-10	44.4240 g (100 mmol)	51.8450 g (100 mmol)	10.6966 g (11.48 mmol)	487.0
6FPAA-15	44.4240 g (100 mmol)	51.8450 g (100 mmol)	16.9886 g (18.24 mmol)	516.0
6FPAA-20	44.4240 g (100 mmol)	51.8450 g (100 mmol)	24.0673 g (25.84 mmol)	548.0
6FPAA-25	44.4240 g (100 mmol)	51.8450 g (100 mmol)	32.0897 g (34.46 mmol)	585.0

**Table 2 nanomaterials-11-01886-t002:** Optical and dielectric properties of 6FPI-POSS composite films.

Samples	*λ*_cut_ ^1^ (nm)	*T*_450_ ^2^ (%)	*T*_450AO_ ^2^ (%)	*n*_AV_ ^4^	*Δn* ^4^	*b** ^3^	Haze (%)	*D* _k_ ^5^	*D* _f_ ^5^
6FPI-0	331	82.8	65.2	1.5792	0.0106	4.40	2.05	2.84	0.009
6FPI-5	326	83.3	0.8	1.5820	0.0110	5.71	2.48	2.76	0.011
6FPI-10	309	84.8	1.9	1.5822	0.0093	5.78	3.27	2.69	0.012
6FPI-15	309	83.1	10.9	1.5806	0.0084	6.28	4.44	2.63	0.012
6FPI-20	294	83.2	25.7	1.5797	0.0110	3.83	4.70	2.58	0.009
6FPI-25	292	84.7	37.1	1.5765	0.0091	3.80	5.14	2.52	0.010

^1^ Cutoff wavelength. ^2^ *T*_450_, *T*_450AO_: Transmittance at 450 nm at a thickness of 25 μm before and after AO erosion; ^3^ yellow index determined by *b**. ^4^ *n*_AV_: average refractive index; Δ*n*: birefringence. ^5^ *D_k_*: dielectric constant measured at 1 MHz; *D_f_*: dielectric loss factor measured at 1 MHz.

**Table 3 nanomaterials-11-01886-t003:** Thermal properties of the 6FPI/POSS composite films.

Samples	*T*_g_ ^1^ (°C)	*T*_5%_ ^1^ (°C)	*T*_10%_ ^1^ (°C)	*R*_w760_ ^1^ (%)	CTE ^1^ (10^−^^6^/K)
6FPI-0	256.2	520	533	53.8	62.3
6FPI-5	258.8	509	525	53.6	64.8
6FPI-10	259.7	494	517	48.4	66.4
6FPI-15	259.3	497	517	48.1	69.4
6FPI-20	259.4	500	517	47.4	69.6
6FPI-25	260.2	494	512	45.3	72.4

^1^ *T*_g_: Glass transition temperatures determined by DSC measurements; *T*_5%_, *T*_10%_: 5% and 10% weight loss temperature, respectively; *R*_w760_: residual weight ratio at 760 °C; CTE: coefficient of linear thermal expansion recorded between 50–200 °C.

**Table 4 nanomaterials-11-01886-t004:** AO effects and erosion yields for 6FPI/POSS composite films.

Samples	TSP-POSS ^1^ (%)	*W*_1_ ^2^ (mg)	*W*_2_ ^2^ (mg)	Δ*W* ^3^ (mg)	*E*_s_ ^4^ (10^−24^ cm^3^/atom)
6FPI-0	0	10.87	4.93	5.94	2.60
6FPI-5	5	12.28	9.06	3.22	1.41
6FPI-10	10	11.86	9.52	2.34	1.02
6FPI-15	15	11.37	9.72	1.65	0.72
6FPI-20	20	11.96	10.74	1.22	0.53
6FPI-25	25	11.87	10.96	0.91	0.40
Kapton^®^	-	30.85	24.00	6.85	3.00

^1^ TSP-POSS content in the PI sample; ^2^ *W*_1_: weight of the sample before irradiation; *W*_2_: weight of the sample after irradiation; ^3^ weight loss of the sample during irradiation, Δ*W* = *W*_1_ − *W*_2_; ^4^ erosion yield.

**Table 5 nanomaterials-11-01886-t005:** XPS results for the AO-exposed and unexposed PI films.

Samples	Relative Atomic Concentration (%)									
	Unexposed Samples					AO-exposed Samples				
	Si2p	C1s	O1s	N1s	F1s	Si2p	C1s	O1s	N1s	F1s
6FPI-0	0	75.74	12.44	2.58	6.58	0	70.81	18.03	2.62	4.54
6FPI-5	5.09	66.58	19.08	3.67	4.39	18.34	35.80	40.43	1.65	3.78
6FPI-10	6.24	67.07	16.07	2.09	8.28	20.08	31.16	44.15	1.14	3.47
6FPI-15	3.14	76.48	13.57	1.64	4.44	17.87	36.74	40.18	1.74	3.47
6FPI-20	2.96	64.63	18.08	3.52	7.56	21.25	28.07	46.51	1.06	3.10
6FPI-25	9.43	66.55	18.74	1.92	3.36	17.30	39.29	38.33	1.84	3.24

## Data Availability

Data is contained within the article.

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
