# Peer review of "Preparation and Characterization of Transparent Polyimide Nanocomposite Films with Potential Applications as Spacecraft Antenna Substrates with Low Dielectric Features and Good Sustainability in Atomic-Oxygen Environments"

_nanomaterials, 2021, doi:10.3390/nano11081886_

Round 1
Reviewer 1 Report
This work presents a lot of experimental results on the study of PI film matrix combined with a nanocage 22 trisilanolphenyl polyhedral oligomeric silsesquioxane (TSP-POSS) additive. The important parameters from the view point of their potential application as spacecraft antenna substrates have been investigated as a function of the amount of TSP-POSS additive. Generally, the work is well written and useful from application point of view. There are some minor issues to be addressed:
- Fig.13 is not discussed in sufficient detail and the discussions at lines 404-412 are not very clear.
- There is a mistake in Table 4 - W2 is obviously the weight AFTER irradiation.
Author Response
Response to reviewer 1:
1. Question: 13 is not discussed in sufficient detail and the discussions at lines 404-412 are not very clear.
Answer: Fig.13 shows the TMA plots of the 6FPI nanocomposite films. We want to express the meaning of influence of TSP-POSS additives on the CTE of the final films. Apparently, incorporation of TSP-POSS filler deteriorated the CTE values of the polymers to some extent. This drawback might be remedied by further modification. We rewrote the discussion in our revised manuscript as follows.
“The pristine 6FPI-0 film itself possessed the CTE value of 62.3×10-6/K due to the presence of flexible hexafluoroisopropylene and ether linkages in the polymer. When the TSP-POSS additives were introduced, the composite films showed further increased CTE values at elevated temperatures. For example, 6FPI-25 exhibited a CTE value of 72.4×10-6/K, which was obviously higher than that of the pristine 6FPI-0 film. Apparently, the plasticization effects of the TSP-POSS fillers increased the CTE values of the composite films. The deteriorated high-temperature dimensional stability might be disadvantageous for their applications as substrates for spacecraft antennas. Modification of the high-temperature dimensional stability of the current PI composite films will be investigated in our future work.”.
2. Questions: There is a mistake in Table 4 - W2 is obviously the weight AFTER irradiation.
Answer: This is our mistake. W2 is indeed the weight after irradiation. We modified it in our revised manuscript.
Reviewer 2 Report
The reviewed manuscript "Preparation and Characterization of Transparent Polyimide Nanocomposite Films with Potential Applications as Spacecraft Antenna Substrates with Low Dielectric Features and Good Sustainability in Atomic-oxygen Environments" by Yan Zhang, Bo-han Wu, Han-li Wang, Hao Wu, Yuan-cheng An, Xin-xin Zhi and Jin-gang Liu, presents interesting research results. The obtained and tested films 6FPI/POSS shows better properties than previously used in antennas in Low Earth Orbit (LEO) satellites. The addition of the TSP-POSS admixture to 6FPI increased the resistance to AO radiation by creating a protective layer on the surface.
The work is written carefully. It is necessary to improve in the following places:
- line 442: below Table 4 is:”W2:Weight of the sample before irradiation”
should be: “W2: Weight of the sample after irradiation”
- line 452: in inequalities omitted sample 6FPI-10-AO
Author Response
Response to reviewer 2:
1. Question: Line 442: below Table 4 is: “W2: Weight of the sample before irradiation” should be: “W2: Weight of the sample after irradiation”
Answer: This is our mistake. W2 is indeed the weight after irradiation. We modified it in our revised manuscript.
2. Question: Line 452: in inequalities omitted sample 6FPI-10-AO
Answer: We have changed “6FPI-0-AO> 6FPI-25-AO> 6FPI-20-AO> 6FPI-15-AO> 6FPI-5-AO” to “6FPI-0-AO> 6FPI-25-AO> 6FPI-20-AO> 6FPI-15-AO> 6FPI-10-AO> 6FPI-5-AO” in our revised manuscript.